# Voltage-controlled magnetoelectric devices for neuromorphic diffusion process

Yang Cheng[1] ✉, Qingyuan Shu[1], Albert Lee[1], Haoran He[1], Ivy Zhu[2], Minzhang Chen[1], Renhe Chen[3], Zirui Wang[4], Hantao Zhang [5], Chih-Yao Wang[6], Shan-Yi Yang[6], Yu-Chen Hsin[6], Cheng-Yi Shih[6], Hsin-Han Lee[6], Ran Cheng [4,5] & Kang L. Wang [1] ✉

Neuromorphic diffusion models have become one of the major breakthroughs in the field of generative artificial intelligence. Unlike discriminative models that have been well developed to tackle classification or regression tasks, diffusion models aim at creating content based upon contexts learned. However, the more complex algorithms of these models result in high computational costs using today's technologies. Here, we develop a spintronic voltage-controlled magnetoelectric memory hardware for the neuromorphic diffusion process. The in-memory computing capability of our spintronic devices goes beyond current Von Neumann architecture, where memory and computing units are separated. Together with the non-volatility of magnetic memory, we can achieve high-speed and low-cost computing, which is desirable for the increasing scale of generative models in the current era. We experimentally demonstrate that the hardware-based true random diffusion process can be implemented for image generation and achieve comparable image quality to software-based training as measured by the Fréchet inception distance (FID) score, achieving ~$10^3$ better energy-per-bit-per-area over traditional hardware.

Diffusion processes are ubiquitous. In human brains, the membrane potential of neurons is affected by stochastic noise[1–3], which can be described by the Langevin equation in neuro field theory (Fig. 1a). Inspired by Langevin dynamics, denoising diffusion probabilistic models (DDPM, or diffusion model)[4] were proposed and have become one of the major breakthroughs in deep learning over the last few years. One crucial aspect of DDPM is the construction of a diffusion process, which involves applying Gaussian noise sequentially to the data until it reaches an isotropic Gaussian distribution (Fig. 1b). While the diffusion model has demonstrated significant potential in applications such as image generation, data recovery, and inpainting, it encounters constraints related to computing speed and energy efficiency. Like other advanced models such as ChatGPT, the parameter space of today's deep learning algorithms has drastically increased

from million to trillion[5] to tackle more and more complex demands. Such large-scale brain-inspired neuromorphic computing cannot be efficiently implemented using the conventional Von Neumann architecture computers[6,7], where storage and computing units are separated, leading to tremendous energy and latency costs in shuttling data back and forth.

Spintronic devices offer a natural solution to the limitations of current computing hardware in supporting neuromorphic computing algorithms. The bits "1" and "0" are represented by the spin-up and spin-down states in magnetic materials that can be integrated with the complementary metal-oxide-semiconductor (CMOS) back-end-of-line (BEOL) process with high-volume production tools. The non-volatility of the magnetic material allows for low energy consumption in compact data storage. The manipulation of the two spin states via an

[1]Department of Electrical and Computer Engineering, University of California, Los Angeles, CA, USA. [2]Department of Physics, The Ohio State University, Columbus, OH, USA. [3]Department of Electrical and Computer Engineering, University of California, San Diego, CA, USA. [4]Department of Electrical and Computer Engineering, University of California, Riverside, CA, USA. [5]Department of Physics and Astronomy, University of California, Riverside, CA, USA. [6]Industrial Technology Research Institute, Taipei, Taiwan, ROC. ✉e-mail: cheng991@g.ucla.edu; wang@ee.ucla.edu

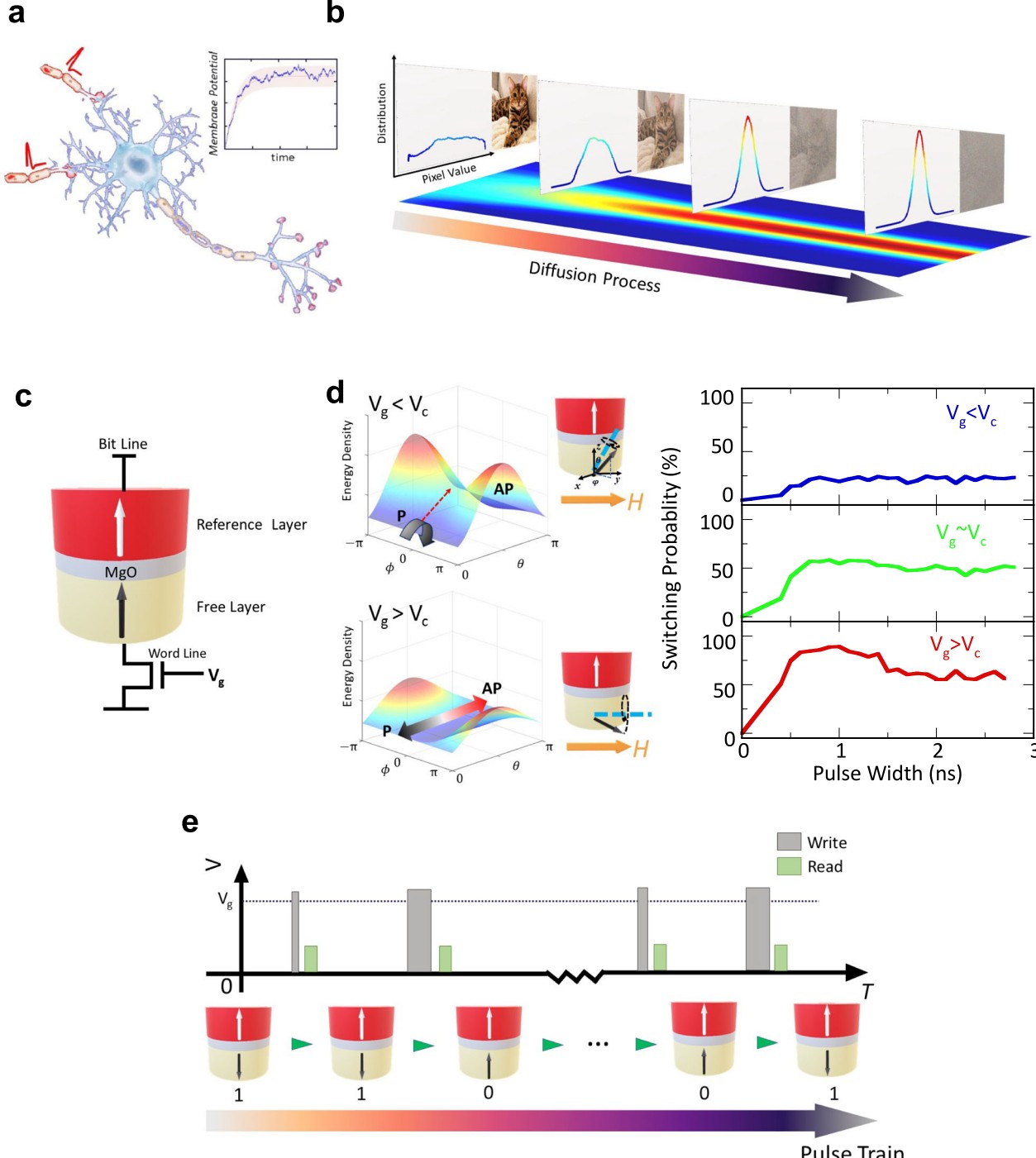

**Fig. 1 | Diffusion process with MTJ. a** Stochastic diffusion process in the membrane potential of the spiking neuron. Membrane potential of a neuron is influenced by various inputs, including synaptic transmissions and inherent noise within the neural system. These inputs cause fluctuations in the membrane potential, leading to a stochastic, or random, component in its behavior. This stochastic nature is crucial for understanding neuronal dynamics, as it affects the likelihood and timing of a neuron firing an action potential; **b** Adding Gaussian noise to an image sequentially until it becomes an isotropic Gaussian distribution. The curves shown represent the distributions of pixel values, which are converted to grayscale from the original RGB channels. **c** Schematic of a MTJ device. Gate voltage is applied between word line and bit line (See Supplementary Information for more details of the construction of a P-cell with MTJ and transistors). **d** (Left) Energy density map of

the MTJ free layer provided by the externally applied field and the PMA modified by VCMA. When the gate voltage is low, the high energy barrier hinders the switching between P and AP states. When the gate voltage is large enough to compensate the PMA ($K_u = 0$ when $V_g = V_c$), switching probability increases. (Right) Numeric simulation of the pulse width dependence of the switching probability under different gate voltage. The switching probability is less than 50% when $V_g < V_c$. When $V_g$ is near and above $V_c$ the switching probability starts to oscillate with the pulse width and will be damped to 50% when the pulse width is long enough. **e**, Diffusion process in MTJ. When a pulse train is sequentially applied to an MTJ, the state of the MTJ is determined by its previous state. The switching probability could be tuned by the width and the voltage magnitude of each write pulse.

electric field or current provides the capability of in-memory computing that goes beyond the conventional Von Neumann computational paradigm. This is particularly ideal for neuromorphic computing, which takes inspiration from the human brain. Some spintronic devices have been proposed as artificial synaptic coupling amongst neurons, nonlinear activation functions, and reservoir layers[8–13]. Simple tasks such as pattern and vowel recognition using a few-layer fully connected neural network or recurrent neural network (RNN) have been demonstrated[14–22]. Compared to the above discriminative models that only draw boundaries by dividing the data space into different classes, generative models that aim at understanding how data are embedded into the space are more difficult to learn but more important for advanced artificial intelligence[23,24]. However, using energy-efficient spintronic devices to tackle the more complex generative tasks has not yet been achieved.

In this article, we report the use of a CMOS-integrated voltage-controlled magnetoelectric random access memory (MeRAM, or VC-MRAM) for the diffusion process. We demonstrate that the switching probability of spin states can be tuned by changing the voltage pulse width and magnitude, via the voltage-controlled magnetic anisotropy (VCMA) effect in our on-chip fabricated magnetic tunneling junctions (MTJ). By sequentially applying a pulse train to an MTJ, the spin state will be updated accordingly and form a Markov chain. Combining multiple such MTJs to an MeRAM array with assigned integer and fraction bit-widths, we can achieve a desirable complex diffusion process. We show that the highly energy efficient MeRAM-based stochastic diffusion process can be successfully implemented into DDPM for image generation. The step-by-step evolution of the MeRAM readout emulates the change of image pixel value under a Markov process in the diffusion model. The quality of generated images matches that of a software-trained model measured by FID score[25].

The building block of data storage for MeRAM is an MTJ where it has two magnetic layers and an MgO insertion in between[26], as shown in Fig. 1c. The magnetization state of the reference layer ($m_{ref}$) is fixed, whereas that of the free layer ($m_{free}$) can be manipulated by the gate voltage applied across the MgO barrier through the VCMA effect. The physical origin of the VCMA effect relates to the modulation of the carrier density at the interface or the electric-field-induced changes of the orbital magnetic moment[27–29]. Figure 1d illustrates how the voltage pulse width and magnitude affect the switching probability of the free layer. Suppose that the initial state is where $m_{free}$ is in parallel with $m_{ref}$ (P state or bit 0) and an in-plane magnetic field is applied, under a small gate voltage $V_g$ ($V_g < V_c$), the VCMA effect does not cancel the perpendicular anisotropy (PMA) in the free layer. This makes the equilibrium position of $m_{free}$ close to the P state. Therefore, there is only a small chance that the $m_{free}$ can be switched to be antiparallel to $m_{ref}$ (AP state or bit 1) due to thermal fluctuations after turning off the pulse. When $V_g$ is large enough to compensate PMA ($V_g > V_c$), the equilibrium position of $m_{free}$ is in plane along the external field direction. The $m_{free}$ undergoes a damped oscillation towards in-plane[26,30]. Then the switching probability also oscillates depending on the relative position of $m_{free}$ when $V_g$ is off. At the first 1 ns, the switching probability increases to approach a near 100% deterministic switching. However, when the pulse is long enough to make $m_{free}$ in-plane, the switching rate would stay around 50% as pure thermal random switching (See Methods for the details of the simulation). Given the deterministic switching at a short pulse width and a 50% switching rate for a long pulse width, MeRAM has been demonstrated as a promising candidate for high energy efficient non-volatile memory (NVM) as well as a true random number generator in probabilistic computing[31–35]. However, its sequential stochastic generation capability has not been explored. In this work, combining the low cost of NVM with its tunable switching rate, we achieve an in-memory Markov process using MeRAM[36]. As shown in Fig. 1e, when a pulse train with different gate voltage and width for each pulse is applied, the state of

MTJ continuously changes and only depends on the previous state and the applied pulse, forming a stochastic diffusion process.

## VC-MTJ properties

We first characterize our MeRAM device consisting of a single MTJ. On-chip MTJs with a diameter of 100 nm are fabricated with the full stack on an 8" wafer shown in Fig. 2a (See Methods for details of fabrication and structure characterization). As the reference layer is pined by a synthetic antiferromagnet layer, the VCMA effect at MgO/CoFeB free layer interface modifies the PMA energy density ($K_u$). (See Supplementary Note 3 for the details about measurements of $K_u$ in the free layer). A VCMA coefficient of 40 fJ/Vm is extracted by a linear fit of $K_u$ to the electric field[37,38]. The read out of MTJ states is made through measuring the tunnel magnetoresistance (TMR) between the bit line and word line. Figure 2b shows the out-of-plane field dependence TMR measurement of our MTJ device. The external field switches the free layer between P and AP states, where the AP state has a high magnetoresistance due to the mismatch of majority and minority spin channels[39]. Our measurement shows a 220% on/off ratio ($R_{AP}/R_P$) between the two states. To demonstrate the tunability of the switching rate, we perform electric-field-induced switching measurements. Figure 2c shows the obtained switching probability using voltage pulses with various lengths and amplitudes. A pulse train is applied to the device and the MTJ states are recorded using a fast oscilloscope, as shown in Fig. 2d, e (See Methods for the experiment setup). Then the switching probability can be extracted by counting the number of AP->P and P->AP states. When $V_g$ is 2.1 V which is below the critical voltage $V_c$ of 2.4 V, a low switching probability is observed. When $V_g = V_c$, the switching probability saturates at 50% as an indication of thermal fluctuation induced stochastic switching (Details of the switching profile at 0.4 ns and 2 ns are shown in Fig. 2d, e). When $V_g$ is 2.7 V which is above $V_c$, voltage-induced precessional motion of magnetization leads to the damped oscillation of switching probability, which eventually ends up as 50%. This is consistent with our numerical simulation as shown in Fig. 1d.

## MeRAM-based Gaussian noise generation

For practical use, we need a MeRAM unit consisting of multiple MTJs, with each pulse train updating the states of the MTJ as a Markov chain. Figure 3a shows a MeRAM unit with eight MTJs connected by the bit line. There are two integer bit-width and six fraction bit-width with assigned digit values. The read out of MeRAM by the post processing unit gives $A_i = \sum_{bit=1}^{8} 2^{(bit-7)} \cdot MTJ_i(bit)$, where $A_i$ ranges from 0 to $\frac{255}{64}$. The distribution of $A_i$ can be obtained by

$$\begin{pmatrix} P_{A_i=0} \\ . \\ . \\ . \\ P_{A_i=\frac{255}{64}} \end{pmatrix} = M \begin{pmatrix} P_{A_{i-1}=0} \\ . \\ . \\ . \\ P_{A_{i-1}=\frac{255}{64}} \end{pmatrix} \quad (1)$$

$$M = \begin{pmatrix} 1-P_{P\to AP}^{(1)} & P_{AP\to P}^{(1)} \\ P_{P\to AP}^{(1)} & 1-P_{AP\to P}^{(1)} \end{pmatrix} \bigotimes \cdots \bigotimes \begin{pmatrix} 1-P_{P\to AP}^{(8)} & P_{AP\to P}^{(8)} \\ P_{P\to AP}^{(8)} & 1-P_{AP\to P}^{(8)} \end{pmatrix} \quad (2)$$

$M$ is the Markov matrix defined by the Kronecker product of eight single transition matrices for each individual MTJ. Further, we need to calculate the distribution of $\varepsilon$, where $\varepsilon(i) = A_i - A_{i-1}$ generated by our MeRAM to match the desired neuromorphic diffusion process. In the diffusion model, $\varepsilon$ is taken as the noise added to the data, which ideally follows a Gaussian distribution (See Supplementary Note 4 for derivation). Working backwards from the target

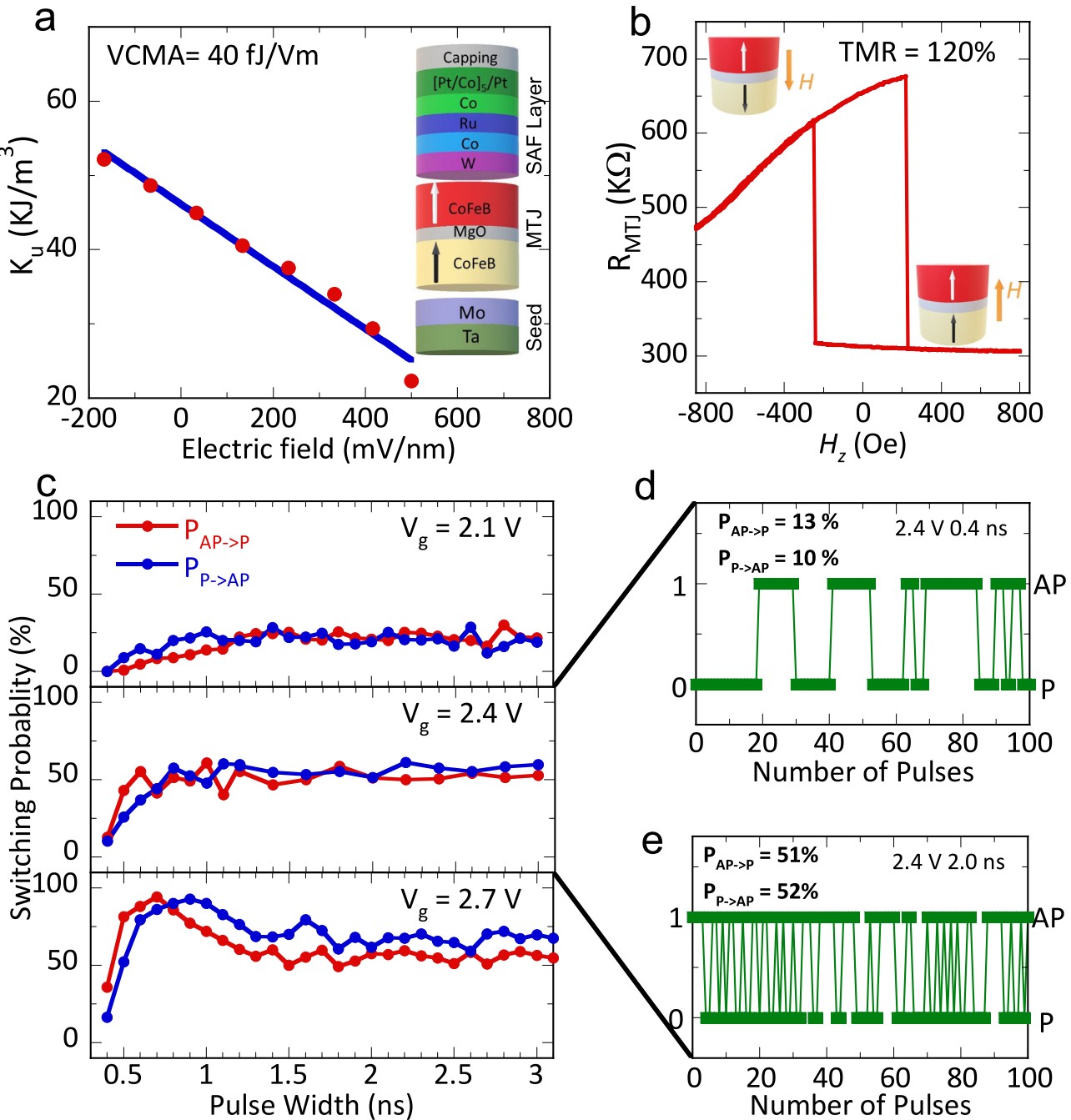

**Fig. 2 | Characterization of voltage-controlled MTJ. a** Measurement of VCMA in MTJ. To VCMA coefficient is extracted by fitting the DC voltage dependent PMA energy. Inset, Schematic of MTJ full stack. **b** TMR measurements. When the out-of-plane field is applied, 120% TMR which corresponds to 220% on/off ratio is achieved. **c** Pulse width dependence of switching probability between P and AP states under gate voltages of 2.1, 2.4, and 2.7 V, respectively. **d** Read out of the MTJ states under a pulse train at 2.4 V and the pulse width of 0.4 ns. We only show 100 pulses (from 1000 pulses) for illustration purpose. **e** Read out of the MTJ states under a pulse train at 2.4 V and the pulse width of 2 ns.

distribution, we can extract the voltage pulse widths and magnitudes on each MTJ in the MeRAM unit. In our experiments, 2.4 V gate voltage is applied with a 0.4 ns pulse width to the leading MTJ (Most Significant Bit, or MSB) while 2 ns pulse for the remaining MTJs. Figure 3b shows the distribution of sampled 10,000 and 40,000 $\varepsilon$ values from the pulse trains. With a larger sample size, the distribution of $\varepsilon$ gets closer to a standard Gaussian as expected. As a proof-of-concept, the 8-bit MeRAM array allows the sampling of standard Gaussian noise in a range of 4$\sigma$. More bits or arrays can increase the range and granularity of the generated noise (See Supplementary Note 4 for details). Compared with a conventional

CMOS-based pseudo random number generator which needs extra bias generators to achieve tunable switching probability, our voltage-controlled MTJ saves 80% of energy and has ~$10^3$ higher figure of merit (FOM) (See Supplementary Note 5), in addition to having true randomness.

## Image generation tasks in CMOS-integrated MeRAM array

With the demonstrated capability of generating Gaussian noise using our VC-MTJ devices, we implement it in DDPM for our image generation task. We first illustrate with a simple letter pattern

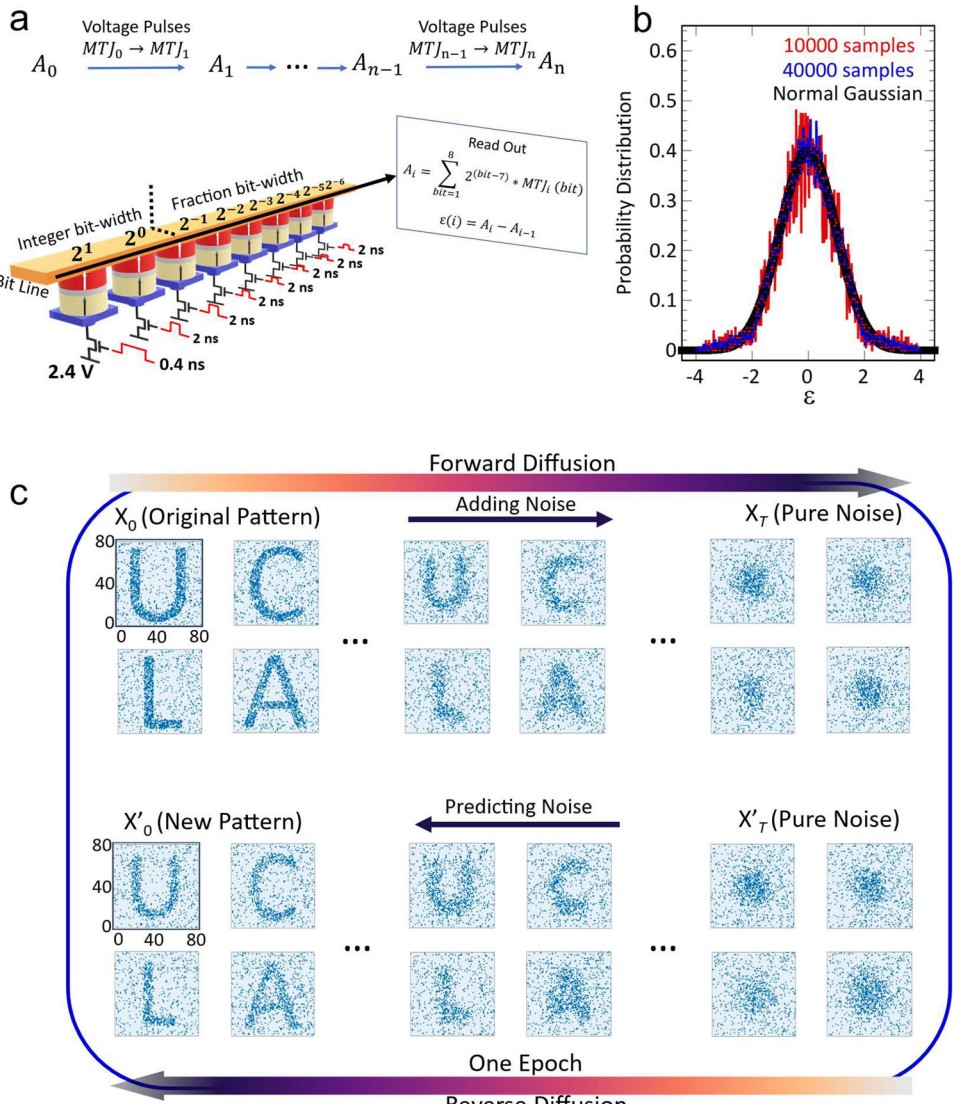

**Fig. 3 | Illustration of MeRAM array for diffusion process. a** An 8-bit MeRAM unit. There are two integer bits and six fraction bits with assigned digit value from $2^1$ to $2^{-6}$. For each operation, 2.4 V gate voltage with distributed pulse widths are applied to each MTJ. **b** Probability distribution of $\varepsilon$ which is defined as the difference between current ($A_i$) and previous ($A_{i-1}$) states. Increasing the sample size makes the distribution close to desired standard Gaussian distribution. **c** Demonstration of generative diffusion processes in a CMOS-integrated MeRAM array made by 80 × 80 MTJs. The training in the forward diffusion process consists of 100 steps. To learn the added noise at each step, we use variational Bayesian methods implemented with a neural network. In the reverse diffusion (generating) process after training, we subtract the learned noise in our MeRAM array step by step, and eventually generate a new pattern aligned with our expectation. Here, dark blue and light blue represent a P state and an AP state, respectively.

learning and generation task using our CMOS-integrated MeRAM array with 80 × 80 VC-MTJs. This marks the first time a MeRAM device has been integrated with the 180 nm CMOS process[40]. As shown in Fig.3c, DDPM contains two parts, a forward diffusion process and a reverse diffusion process. In the forward diffusion process, Gaussian noises are added to the training data (pixel data $\mathbf{x_0}$) in $T$ steps sequentially, with the variance increasing with each step $t$. When $T$ is large enough, $\mathbf{x_T}$ is subject to a Gaussian distribution. In the reverse diffusion process, starting from pure Gaussian noise $\mathbf{x_T}$, pixel distribution $\mathbf{x_{t-1}}$ is learned by comparing the known posterior distribution $q(\mathbf{x_{t-1}}|\mathbf{x_t})$ to the predicted distribution $p_\theta(\mathbf{x_{t-1}}|\mathbf{x_t})$. Using the variational inference method, the Kullback–Leibler divergence (KL-divergence) of $q(\mathbf{x_{t-1}},|,\mathbf{x_t})$ and $p_\theta(\mathbf{x_{t-1}}|\mathbf{x_t})$ simplifies to the prediction of the noise distribution from $\mathbf{x_t}$ to $\mathbf{x_{t-1}}$. This can be achieved by introducing a convolutional neural network-based U-Net. We repeat the above process in one epoch and perform multiple iterations (epochs) to improve the performance of the diffusion model. To generate images, we use the trained noise in the reverse diffusion process to sample a new denoised image $\mathbf{x'_0}$ step by step from $\mathbf{x_T}$. In our experiment, we choose to use a total step $T$ of 1000. In our MeRAM unit, we utilize a set of 40,000 random numbers sampled from our device as the noise dataset, applying it in both the training phase (Forward Diffusion) and the generation phase (Reverse Diffusion). This involves changing the states of VC-MTJs by applying voltage to the MeRAM array. In the images, each pixel corresponds to one VC-MTJ, with dark blue and light blue representing a P state and an AP state, respectively. Here, $\mathbf{x_t}$ is the coordinate of P states, and the trajectories of P states (change from $\mathbf{x_t}$ to $\mathbf{x_{t+1}}$) follow a Gaussian diffusion process[41]. We have separately trained the array on different letter patterns such as "U", "C", "L", and "A", and subsequently generated new patterns using the trained models. The results align with our expectations, showcasing the effective pattern generation capabilities of our device.

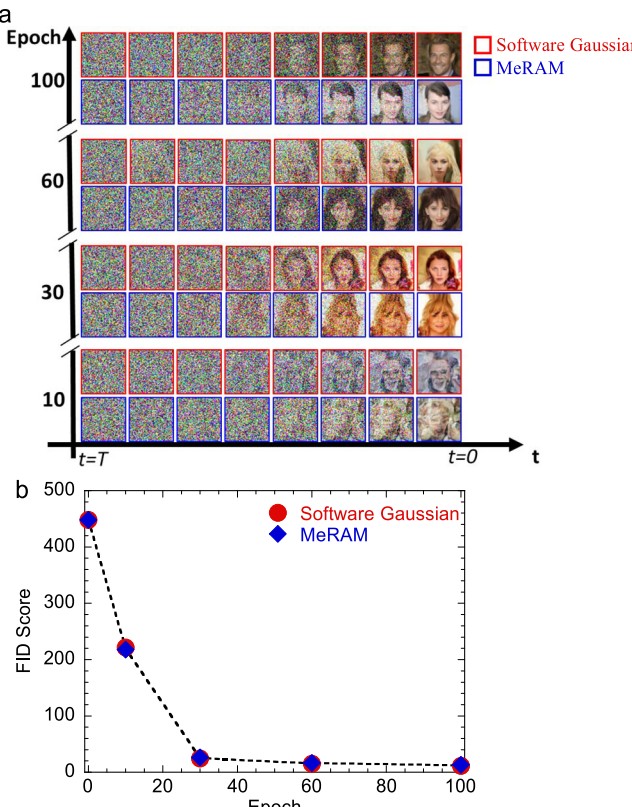

**Fig. 4 | Performance of imaging generation task using MeRAM based diffusion model. a** Evolution of image generation using software-based and MeRAM-based diffusion processes with increasing training epochs. **b** The quality of generated images measured by the FID score, where a lower score corresponds to a better quality. After 100 epochs of training, the FID score for our MeRAM-based generation stands at 13.1, closely aligning with the 12.1 FID score achieved by software-generated Gaussian noise.

## Performance of MeRAM-based generative diffusion model

To evaluate the performance of our MeRAM-based generative diffusion model, we utilize the CelebFaces Attributes (CelebA-HQ) dataset[42] with $64 \times 64$ resolution images for training. We follow the original DDPM setup, where noise is added to each pixel value ($\mathbf{x}_t$) to learn the connections prior to image generation. Figure 4a illustrates the image generation process across different training epochs. Notably, after 100 epochs, both the MeRAM-based and the fully software-based diffusion models produce high-quality images. We quantify image quality using the FID score, where lower scores indicate better quality. We compare the images generated using the software-based and our MeRAM-based diffusion process across various numbers of training epochs (See Supplementary Information for details of DDPM implementation). As shown in Fig. 4b, despite the slight deviations of the MeRAM-generated noise from a strictly Gaussian distribution, the improvements in image quality with an increasing number of epochs occur at the same rate for both the software-based and MeRAM-based diffusion models, but with much lower energy consumption.

## Discussion

In conclusion, we have demonstrated the use of voltage-controlled MeRAM for the neuromorphic diffusion process. Notably, we have successfully accomplished high-quality image generation using an integrated CMOS MeRAM chip for the first time. Our work goes beyond traditional discriminative models, implementing MeRAM to advance the state-of-the-art generative diffusion models. In the

context of DDPM, the precise and controlled generation of noise is a central component. By enhancing noise generation efficiency using MeRAM, we are able to reduce the iterative burden typically associated with these models, leading to a much more streamlined and energy-efficient process. Furthermore, MeRAM overcomes the limitations of conventional STT-based MRAM, which faces higher energy consumption, limited endurance, and reduced speed due to incubation delay[43]. Additionally, while STT and SOT-MRAM requires a low energy barrier to function as a probabilistic bit (p-bit)−compromising retention time −VCMA-based MTJ devices in offer the unique flexibility to serve as both p-bits and memory cells, making it particularly suitable for large-scale neuromorphic probabilistic computing applications[44,45]. We believe our spintronics hardware could overcome one of the biggest challenges in the current era of neuromorphic computing−the gap between the increasing complexity of algorithms and the computing platform that remains in von Neumann architecture.

## Methods

### Fabrication and characterization of MTJ

Magnetic multilayer stacks are grown on an 8-inch CMOS backend wafer by sputtering, followed by annealing at 360 °C for 20 min, which is compatible with standard CMOS backend processing. MTJ is defined by E-beam lithography and its diameter is 100 nm. The standard MTJ fabrication process is performed on the whole wafer. Transmission electron microscopy and scanning electron microscope are used to characterize the growth and fabrication of the MTJ stack, as shown in Supplementary Fig. 1. For the switching probability measurements, we use a GMW 5201 Projected Field Electromagnet driven by a Kepco Bipolar Operational Power Supply to apply an external in-plane magnetic field 390 Oe. A Keithley 2636A source meter is used for (a) triggering a Tektronix PSPL10050A Programmable Pulse Generator to generate voltage pulses over the MTJ device; (b) applying a small constant voltage across the MTJ for resistance readout. This voltage is applied across a constant series resistance that serves as a voltage divider and is injected to the MTJ through the DC input of a Bias Tee. To collect the data, an Agilent MSO7014B oscilloscope is connected across the MTJ. The switching behavior is reflected by the voltage fluctuation recorded by the oscilloscope.

### Simulation of switching probability under different voltage

Macrospin simulations are performed to obtain the switching probability in the thermal activation regime, namely, when the voltage is slightly below the critical voltage that removes the barrier completely. In the simulation, we numerically integrate the Landau-Lifshitz-Gilbert equation using Matlab

$$\dot{\boldsymbol{m}} = -\gamma_0 \boldsymbol{m} \times (\boldsymbol{H}_{eff} + \boldsymbol{h}) + \alpha \boldsymbol{m} \times \dot{\boldsymbol{m}} \qquad (3)$$

where the effective field $\boldsymbol{H}_{eff} = -\partial_{\boldsymbol{m}} w / \mu_0 M_s$ is obtained from the gradient of the energy density profile $w$ containing the PMA energy and Zeeman energy

$$w = K_{eff}(V)(1 - m_z^2) - \mu_0 M_s H_x m_x \qquad (4)$$

The effective anisotropy $K_{eff}(V)$ is controlled by the applied voltage across MgO via the VCMA effect. For sufficiently large $K_{eff}$, the energy landscape has two energy minima corresponding to the two PMA states. To investigate the switching probability, we include a white noise as the thermal random field $\boldsymbol{h}$. Following Brown's derivation[46],

$$\langle h_i(t) \rangle = 0, \ \langle h_i(t) h_j(t + \tau) \rangle = \mu \delta_{ij} \delta(\tau) \qquad (5)$$

where $\mu = \frac{2 k_B T \alpha}{\gamma_0 \mu_0 M_s V}$ is used to satisfy the thermal equilibrium condition. Numerically, we follow Scholz's approach and use the 2nd order

Heun's method to integrate the dynamics[47]. Adopting this approach ensures a good balance between numerical stability and complexity.

During the simulation, we initialize the spin in the absence of an external voltage so that the state will evolve from one of the two strong PMA states. Then, we turn on a voltage pulse with sharp edges. The PMA energy term is instantly modified, thereby affecting the dynamics through the effective field. The switching event is determined if the state is trapped in the other PMA state a few nano-second after the voltage is turned off. The switching probability $((P_{AP->P} + P_{P->AP})/2)$ is obtained by counting the number of the switching events among 150 trials.

## Data availability

All data needed to evaluate the conclusions in the paper are available within the article and its Supplementary Information files. All data generated during the current study are available from the corresponding author upon request.

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

## Acknowledgements
The authors in University of California, Los Angeles acknowledge the support from the National Science Foundation (NSF) Award No. 1810163 and No. 2427172; This work at University of California, Riverside is supported by the Air Force Office of Scientific Research under Grant No. FA9550-19-1-0307.

## Author contributions
Y.C. designed, planned and initiated the study. Q.S., H.H., and Y.C. performed the voltage-controlled switching probability measurement. A.L., R.H.C. and Z.W. performed circuit implementation simulations. I.Z. performed the DDPM training. C.Y.W., S.Y.Y., Y.C.H., C.Y.S. and H.H.L. grew and fabricated devices. H.Z. and R.C. contributed to the theoretical modeling of Markov process. Y.C. and K.L.W. supervised the project. Y.C., Q.S., A.L., H.H., I.Z., M.C., R.H.C., Z.W., and K.L.W. drafted the manuscript. All authors discussed the results and commented on the manuscript.

## Competing interests
The authors declare no competing interests.
