## [Peer Review File · Nature Communications]

Voltage-Controlled Magnetoelectric Devices for Neuromorphic Diffusion Process

Corresponding Author: Professor Kang Wang

Version 0:

Reviewer comments:

Reviewer #1

(Remarks to the Author)

The manuscript entitled Voltage-Controlled Magnetoelectric Devices for Neuromorphic Diffusion Process by Y. Cheng et al describes the use of CMOS-integrated voltage-controlled magnetoelectric random access memories (MeRAM) for implementing a neuromorphic diffusion process. The authors claim that the developed system has a superior performance than current semiconductor solutions, exhibiting a low energy consumption and fast response (resistance switching of the MTJs occurs for 4 ns pulses).

Neuromorphic computing is a hot topic demanding devices capable to implement new computing concepts. Spintronic devices, particularly Magnetic Random Access Memories based on Magnetic Tunnel Junctions, have been proposed as a main candidate to implement new memory devices, like low consumption ultra-fast memories, or memristive devices, spiking neurons among others. Low energy applications of spintronic devices are of high interest since the use of conventional CMOS technology needs an elevated consume of energy. Voltage-controlled MTJs results in a low energy consumption option capable to implement new device concepts.

I truly believe that this work will attract the attention of two kind of audiences, researchers working in materials and devices and groups oriented in the development of systems for neuromorphics. The group has a long experience in magnetic tunnel junctions, what is clearly seen in the manuscript offering a detailed characterization and integration of the MTJs with CMOS technology. I consider myself from the first ones, and for this reason I miss some more details explaining the link between the device and the circuit to implement the diffusion model.

- The authors claim that VCMA devices show greater performance than other devices driven by spin torques. This is shown in figure 6 with a graph comparing the writing speed of MTJs with size. Can the authors give some detailed parameters and figures of merit to show their claim? In my opinion, figure 6 is a clear comparison for example for memory devices, but I am not sure if captures the essential main differences to integrate a diffusion process.
- The proposed MeRAM use a line of 8-bit MeRAM to generate random numbers. The authors explain that is possible to get a faster result with more bits. What is the minimum number of bits to get diffusion process in a reasonable amount of time? How the energy consumption of the system increases with the number of bits?

The paper is well written and easy to follow. I think that the authors need to link better the MTJ performance with the characteristics of their neuromorphic computing. I cannot recommend their work for publication at this point.

Reviewer #2

(Remarks to the Author)

The authors demonstrate the use of voltage control magnetic anisotropy/magnetoelectric magnetic tunnel junction (MTJs) for neuromorphic computing applications, in particular for the denoising diffusion probabilistic model used in deep learning. The manuscript is clear and well-organized. However, the work described here has not sufficiently discussed and referenced its novelty and significance against similar MTJ-based neuromorphic work such as in D. Zhang et al. IEEE transactions on biomedical circuits and systems 10, 828 (2016); P. Rzeszut et al. Scientific Reports 12, 7178 (2022); E. Raymenants et al. Journal of Applied Physics 124, 152116 (2018); A. Kumar et al. ACS Applied Materials and Interfaces 16, 10335 (2024); A.

Further comments are listed as follows

- 1) Fig. 2b: The antiparallel state of the MTJ fluctuates by ~15% with the magnetic field, which is highly unstable. Do all devices exhibit this kind of behavior? Could the author clarify why this state isn't stable and how this will affect the performance of the image generation task?
- 2) Fig. 2c: What is the rise time of the pulse generator used in this experiment? The sub-ns pulses can become nearly triangular in this range. Have the authors performed any calibration using an oscilloscope or any other technique?
- 3) Fig. 2c: The VCMA effect often oscillates with an increase in pulse width. The switching probability may be measured for some intermediate voltages and larger pulse widths to verify the trend and period of oscillation, if any.
- 4) Fig. 2d,e: Shao et al. 3, 87 Communications Materials (2022) showed sub-nanosecond precessional VCMA (>100 fJ/Vm) assisted switching of 50 nm MTJs using a voltage lower than 1 V. The authors should benchmark the parameters of their VC-MTJ devices, and implications on the diffusion process and neuromorphic computing performance, against other VC-MTJ reported in the literature such as Communications Materials (2022), AIP Advances 7, 055934 (2017), Appl. Phys. Lett. 108, 012403 (2016), Appl. Phys. Express 9, 013001 (2016) and IEEE Transactions On Nanotechnology 22, 2023, etc.
- 5) Fig:2c-e: Is there any in-plane assistance field applied during measurements? Processional switching may necessitate the use of an in-plane field.
- 6) Authors should provide a detailed documentation of the pulsing parameters and simulation set up and software in the Method section
- 7) Can the author quantify the thermal stability factor or energy barrier for the fabricated devices?
- 8) Is 8 MTJs the optimal arrangement? How does the range and granularity of the noise as a function of number of MTJs affect the FID?
- 9) Authors should include a schematic and/or images of the 80x80 VC-MTJ array and circuitry to help readers visualize their setup. Additionally, to show supporting data such as cross-sectional TEM of the integrated MRAM devices on CMOS wafer given the claim that it is the first time it has been achieved.
- 10) The author should comment whether the SOT/STT MTJ, without the VC mechanism, can perform the DDPM. Additional FID simulations should be performed for SOT/STT MTJ, without VC, to substantiate the high-performance image generation capability, if any, of the VC mechanism over SOT/STT-MTJ.

Reviewer #3

(Remarks to the Author)

Reviewer #4

(Remarks to the Author)

Stochastic diffusion processes, driven by random number sequences, are highly valuable in neuromorphic computing and generative artificial intelligence (AI). However, conventional computers face significant inefficiencies in running AI models based on diffusion processes, primarily due to the separation of storage and computation, as well as the challenges of generating random numbers efficiently. Given the increasing demand for generative AI computing, there is a strong need for new nano-electronic devices that incorporate non-volatility and intrinsic randomness. In this manuscript, the authors present spin-based hardware for Gaussian noise generation and diffusion process. The switching probabilities have been adjusted by varying the voltage pulse width and magnitude in voltage-controlled magnetic tunneling junctions (VC-MTJs) under an external magnetic field. Additionally, they demonstrate image generation tasks using CMOS-integrated voltage-controlled magnetoelectric random access memory (VC-MRAM). While their findings offer a possible approach to efficient Gaussian noise generation and diffusion process, there are some key issues that need to be clarified.

1. Limited tunability of the VC-MTJ switching probability. In Figure 2c, it is unclear why the switching probability does not vary smoothly with the voltage pulse width as expected, and why the PAP→P and PP→AP are not equal. Moreover, achieving a time resolution as precise as 0.1 ns is challenging in practical circuits, making it difficult to accurately control the switching probability of the VC-MTJ. Do these limitations affect the quality of Gaussian noise generation and the diffusion process? This requires further clarification.
2. Uncertainty in the robustness of VC-MTJ probabilistic properties. It remains unclear how many times the VC-MTJ can be reliably controlled by voltage pulses without significant changes in switching probability. The manuscript does not address how the switching probability evolves after repeated application of identical pulses to a single VC-MTJ, which is critical for evaluating the device's lifetime in probabilistic applications. This issue needs to be clarified.

3. Device-to-device variability in probabilistic properties. The manuscript does not address the variability in probabilistic behavior across different devices in the VC-MRAM array. For instance, what is the distribution of switching probabilities within the VC-MRAM array when identical 2.4V@2ns voltage pulses are applied? If the variability between devices is significant, how does this impact the accuracy of Gaussian noise generation? Furthermore, how should this variability be calibrated to ensure consistent performance?
4. Unclear role of VC-MRAM in denoising diffusion probabilistic models (DDPM). The manuscript does not clearly explain how the noisy dataset generated by VC-MRAM is used during the image generation phase (reverse diffusion process) of the DDPM. How is the switching probability of each VC-MTJ determined in each iteration? It would be helpful to include a detailed explanation of at least one iteration of the specific calculations involved in the reverse diffusion process.
5. Requirement for an external magnetic field. The need for an external magnetic field is not conducive to practical applications, as its high energy consumption undermines the goal of low-power applications. Additionally, it raises the question of whether a slanted magnetic field affects the switching probability of the VC-MTJ. This should be addressed.

Minor issues:

1. What is the critical switching voltage for spin-transfer torque (STT)-induced free-layer magnetization switching in the presence of in-plane magnetic fields? Additionally, why is the contribution of STT omitted in device simulations?
2. In Figure 2a, why does the data point at 500 mV/nm appear significantly offset from the fitted line?
3. The widths of the input voltage pulses applied to the 8-bit VC-MTJ array shown in Figure 3a do not correspond to those described on page 8, line 3-4 of the main text.
4. The formula for A_i in the penultimate line of the second paragraph on page 7 is incorrect.
5. Recently, non-volatile field-free spin-orbit torque (SOT) devices have been demonstrated for probabilistic bits. These studies should be considered when comparing VC-MRAMs with SOT-MRAMs: [Nano Lett. 24, 5420-5428 (2024); Nano Lett. 24, 10072-10080 (2024)].

Version 1:

Reviewer comments:

Reviewer #1

(Remarks to the Author)

The authors address most of my comments and expand their work accordingly. Although the device development part is not accurate as the diffusion method, I think that one of the value of their paper is the implementation of the computing system. I recommend their work for publication in Nature Communications

Reviewer #2

(Remarks to the Author)

The authors have addressed most of the questions satisfactorily by and large and we deem the manuscript to be technically sound. However, there is still a concern on the novelty aspect of this work. The concept of stochastic/probabilistic computing using MRAMs have been well demonstrated (10.1109/NANO61778.2024.10628878, 10.1109/JXCDC.2022.3231550), wherein SOT-MRAM and voltage assisted/MeRAM have been shown to be similarly suited for these systems with small delay and low energy. The use of MeRAM for probabilistic computing is not new either (10.1088/1361-6528/acf6c7, 10.1109/NANO61778.2024.10628878). In this regard, we feel that the significance of this work is not strong and tends towards incremental.

Reviewer #3

(Remarks to the Author)

Reviewer #4

(Remarks to the Author)

I believe the manuscript has been significantly improved with these clarifications and additions. The authors have addressed the key concerns raised in my review, and I now feel that the manuscript provides a more comprehensive understanding of the proposed spin-based hardware for Gaussian noise generation and diffusion processes. I am confident that the revisions will enhance the quality and applicability of this work. I am happy with the current format of the paper.

Response to Reviews on Manuscript NCOMMS-24-53635-T

We thank the four Reviewers for their careful and prompt reviews. Below, we address the Reviewers' questions and comments on a point-by-point basis. We also revised the manuscript and highlighted the changes in blue.

Response to Reviewer #1

The manuscript entitled Voltage-Controlled Magnetoelectric Devices for Neuromorphic Diffusion Process by Y. Cheng et al describes the use of CMOS-integrated voltage-controlled magnetoelectric random access memories (MeRAM) for implementing a neuromorphic diffusion process. The authors claim that the developed system has a superior performance than current semiconductor solutions, exhibiting a low energy consumption and fast response (resistance switching of the MTJs occurs for 4 ns pulses).

Neuromorphic computing is a hot topic demanding devices capable to implement new computing concepts. Spintronic devices, particularly Magnetic Random Access Memories based on Magnetic Tunnel Junctions, have been proposed as a main candidate to implement new memory devices, like low consumption ultra-fast memories, or memristive devices, spiking neurons among others. Low energy applications of spintronic devices are of high interest since the use of conventional CMOS technology needs an elevate consume of energy. Voltage-controlled MTJs results in a low energy consumption option capable to implement new device concepts.

I truly believe that this work will attract the attention of two kind of audiences, researchers working in materials and devices and groups oriented in the development of systems for neuromorphics. The group has a long experience in magnetic tunnel junctions, what is clearly seen in the manuscript offering a detailed characterization and integration of the MTJs with CMOS technology. I consider myself from the first ones, and for this reason I miss some more details explaining the link between the device and the circuit to implement the diffusion model.

Response: Thank you for your positive feedback and thoughtful comments. We have modified our manuscript based on your suggestions, and we are grateful for your insights, which have helped us improve the quality of our paper significantly.

1. The authors claim that VCMA devices show greater performance than other devices driven by spin torques. This is shown in figure 6 with a graph comparing the writing speed of MTJs with size. Can the authors give some detailed parameters and figures of merit to show their claim? In my opinion, figure 6 is a clear comparrision for example for memory devices, but I am not sure if captures the essential main differences to integrate a diffusion process.

Response: Thank you for your question. We acknowledge that Figure S6 primarily focuses on cell size and write speed, which, while important, do not fully capture the broader comparison needed to evaluate performance across different memory technologies. To address this, we have updated the figure into a more comprehensive table that includes key parameters for both non-volatile memories (e.g., MeRAM, STT-MRAM, SOT-MRAM, RRAM, FeFET) and volatile memories (e.g., SRAM, DRAM) with additional references. The table includes key parameters such as cell size, write energy, write speed, retention power, and endurance. These metrics

highlight the specific advantages of VCMA-driven MeRAM, including its superior write speed, energy efficiency, endurance, and compact size, compared to other spin-torque-based and non-volatile memory technologies. While it is true that other non-volatile memory technologies, such as RRAM or STT-MRAM, could also be used for generating random noise in the diffusion process, MeRAM offers advantages that make it more suitable for this application. Additionally, utilizing STT(SOT)-MRAM as a probabilistic bit (p-bit) requires a low energy barrier, which significantly compromises retention time due to reduced perpendicular magnetic anisotropy (PMA). Conversely, using STT-MRAM as memory necessitates a higher barrier, which diminishes the tunability of switching probability. In contrast, the tunability of VCMA-based MTJ devices allows them to function effectively as both p-bits and memory cells, providing flexibility that is unattainable with STT-MRAM. This dual capability is crucial for neuromorphic computing applications, where devices must balance stochasticity and data storage.

We have revised the **Supplementary Information**

	SRAM 19,20	4T GC- eDRAM 21,22	MeRAM	FeFET 23,24	RRAM 25,26	STT- MRAM ²⁷⁻²⁹	SOT- MRAM 27,30-32
Cell Size (F ²)	415	293	16	50	100	16	24
Write Energy (fJ/bit)	0.24	0.33	6	5	10 ² -10 ⁵	120	220
Write Speed (ns)	1.4	0.6	0.5	10	100	10-20	0.5
Non-volatility	No	No	Yes	Yes	Yes	Yes	Yes
Retention Power (pW/bit)	~9.3	~7.2	0	0	0	0	0
Endurance	Infinite	Infinite	> 10 ¹¹	>10 ⁴	>10 ⁶	>10 ¹⁰	>10 ¹²

Supplementary Table 3. Performance of MeRAM compared with other volatile and non-volatile memories.

Reference List

- 19 Ishii, Y. et al. in 2010 Symposium on VLSI Circuits. 99-100 (IEEE).
- 20 Ishii, Y. et al. in 2012 IEEE International Solid-State Circuits Conference. 236-238 (IEEE).
- 21 Giterman, R., Fish, A., Burg, A. & Teman, A. A 4-transistor nMOS-only logic-compatible gain-cell embedded DRAM with over 1.6-ms retention time at 700 mV in 28-nm FD-SOI. IEEE

Transactions on Circuits and Systems I: Regular Papers 65, 1245-1256 (2017).

22 Frankel, B., Sarfati, E., Rossi, D. & Wimer, S. Energy efficiency of opportunistic refreshing for Gain-Cell Embedded DRAM. IEEE Transactions on Circuits and Systems I: Regular Papers 70, 1605-1612 (2022).

23 Trentzsch, M. et al. in 2016 IEEE International Electron Devices Meeting (IEDM). 11.15. 11-11.15. 14 (IEEE).

24 Dünkel, S. et al. in 2017 IEEE International Electron Devices Meeting (IEDM). 19.17. 11-19.17. 14 (IEEE).

25 Wan, W. et al. A compute-in-memory chip based on resistive random-access memory. Nature 608, 504-512 (2022).

26 Sassine, G. et al. in 2018 IEEE International Reliability Physics Symposium (IRPS). P-MY. 2-1-P-MY. 2-5 (IEEE).

27 Pan, C. & Naeemi, A. Nonvolatile spintronic memory array performance benchmarking based on three-terminal memory cell. IEEE Journal on Exploratory Solid-State Computational Devices and Circuits 3, 10-17 (2017).

28 Jan, G. et al. in 2018 IEEE Symposium on VLSI Technology. 65-66 (IEEE).

29 Dong, Q. et al. in 2018 IEEE International Solid-State Circuits Conference-(ISSCC). 480-482 (IEEE).

30 Garello, K. et al. in 2019 Symposium on VLSI Circuits. T194-T195 (IEEE).

31 Natsui, M. et al. Dual-port SOT-MRAM achieving 90-MHz read and 60-MHz write operations under field-assistance-free condition. IEEE Journal of Solid-State Circuits 56, 1116-1128 (2020).

32 Cai, K. et al. in 2022 International Electron Devices Meeting (IEDM). 36.32. 31-36.32. 34 (IEEE).

2. *This he proposed MeRAM use a line of 8-bit MeRAM to generate random numbers. The authors explain that is possible to get a faster result with more bits. What is the minimum number of bits to get diffusion process in a reasonable amount of time? How the energy consumption of the system increases with the number of bits?*

Response: Thank you for this insightful question. We agree that increasing the number of bits in the MeRAM array improves diffusion results. Based on your suggestion, we conducted additional tests, as shown in the following figure. Our results indicate that using 4 to 6 bits can generate images with reasonably good quality for the neuromorphic diffusion process. Additionally, we observed that the energy consumption increases linearly with the number of bits in the array, aligning with our expectations for scalability.

We have revised the **Supplementary Information**.

Supplementary Figure 4. a, FID score as a function of the number of bits in the MeRAM array, ranging from 2 to 8 bits. The FID score stabilizes beyond 6 bits, and using 4-bit precision already produces relatively good image quality.

The paper is well written and easy to follow. I think that the authors need to link better the MTJ performance with the characteristics of their neuromorphic computing. I cannot recommend their work for publication at this point.

Response: We are grateful for your comments, which have guided us in improving the manuscript, and we hope that the revisions now address your concerns.

Response to Reviewer #2

1. The authors demonstrate the use of voltage control magnetic anisotropy/magnetoelectric magnetic tunnel junction (MTJs) for neuromorphic computing applications, in particular for the denoising diffusion probabilistic model used in deep learning. The manuscript is clear and well-organized. However, the work described here has not sufficiently discussed and referenced its novelty and significance against similar MTJ-based neuromorphic work such as in D. Zhang et al. IEEE transactions on biomedical circuits and systems 10, 828 (2016); P. Rzeszut et al. Scientific Reports 12, 7178 (2022); E. Raymenants et al. Journal of Applied Physics 124, 152116 (2018); A. Kumar et al. ACS Applied Materials and Interfaces 16, 10335 (2024); A. Kumar et al. Nanoscale Horizons 9, 1522 (2024).

Response: Thank you for your valuable feedback and for highlighting these important references. We recognize that the works you mentioned are pioneering contributions to MTJ-based

neuromorphic applications, and we agree that they should be referenced in our manuscript. The key novelty of our work lies in utilizing voltage-controlled magnetic anisotropy (VCMA) in our MeRAM, which provides significant advantages over conventional spin-transfer torque (STT)-based MRAM technologies presented in prior pioneering studies. STT-MRAM faces challenges such as high energy consumption, limited endurance, and reduced speed due to incubation delay. Additionally, utilizing STT-MRAM as a probabilistic bit (p-bit) requires a low energy barrier, which significantly compromises retention time due to reduced perpendicular magnetic anisotropy (PMA). Conversely, using STT-MRAM as memory necessitates a higher barrier, which diminishes the tunability of switching probability. In contrast, the tunability of VCMA-based MTJ devices allows them to function effectively as both p-bits and memory cells, providing flexibility that is unattainable with STT-MRAM. This dual capability is crucial for neuromorphic computing applications, where devices must balance stochasticity and data storage.

We have revised our wording to include and acknowledge these studies, discussing how our work builds upon and differs from these foundational efforts.

The references are added at

Some spintronic devices have been proposed as artificial synaptic coupling amongst neurons, nonlinear activation functions, and reservoir layers. Simple tasks such as pattern and vowel recognition using a few-layer fully connected neural network or recurrent neural network (RNN) have been demonstrated.

In Discussion: (More details are listed in **Supplementary Note 6** and **Supplementary Table 3**.) Furthermore, MeRAM overcomes the limitations of conventional STT-based MRAM, which faces higher energy consumption, limited endurance, and reduced speed due to incubation delay. Additionally, while STT and SOT-MRAM requires a low energy barrier to function as a probabilistic bit (p-bit)—compromising retention time—VCMA-based MTJ devices in offer the unique flexibility to serve as both p-bits and memory cells, making it particularly suitable for large-scale neuromorphic probabilistic computing applications.

Further comments are listed as follows :

- 2. Fig. 2b: The antiparallel state of the MTJ fluctuates by ~15% with the magnetic field, which is highly unstable. Do all devices exhibit this kind of behavior? Could the author clarify why this state isn't stable and how this will affect the performance of the image generation task?*

Response: Thank you for pointing this out. In Fig. 2b, we show the magnetic hysteresis loop under a perpendicular magnetic field (originally denoted as H) to extract the TMR of the MTJ. The ~15% change observed at -800 Oe is due to the depinning of the fixed layer under a large perpendicular field, a phenomenon similar to what is reported in Fig. 1d of *A. Kumar et al., ACS Applied Materials and Interfaces 16, 10335 (2024)*. To avoid confusion, we have updated the notation to Hz to clarify the perpendicular field direction. This fluctuation should not impact the performance of the image generation task, as our VCMA switching requires only a small fixed in-plane field, with no need for a perpendicular field.

In Fig. 2b

Fig. 2| Characterization of voltage-controlled MTJ. b, TMR measurements. When the out-of-plane field is applied, 120% TMR which corresponds to 220% on/off ratio is achieved.

3. *Fig. 2c: What is the rise time of the pulse generator used in this experiment? The sub-ns pulses can become nearly triangular in this range. Have the authors performed any calibration using an oscilloscope or any other technique?*

Response: Thank you for your question. We used a Tektronix TDS6145C oscilloscope to capture the voltage pulse shape of 2.1 V. While we observed slight deformation in the pulse shape at the shortest pulse width of 0.4 ns, the overall generation of the pulse shape remains good and sufficient for our experiments.

We have added the **Supplementary Figure 8** as shown below

Supplementary Figure 8. Nanosecond voltage pulse shape calibration. Voltage pulses of 2.1 V captured by oscilloscope.

4. Fig. 2c: The VCMA effect often oscillates with an increase in pulse width. The switching probability may be measured for some intermediate voltages and larger pulse widths to verify the trend and period of oscillation, if any.

Response: Thank you for your insightful comment. Regarding the voltage intervals, unfortunately, our current pulser has a limitation of 0.3V intervals, so we are unable to gather finer data points. As for the oscillation in switching probability with pulse width, we agree that the VCMA effect can exhibit oscillations in switching probability as the pulse width increases. However, in our main experiment, the strong perpendicular magnetic anisotropy (PMA) of our devices, combined with a relatively modest VCMA coefficient, prevented the system from entering a full oscillatory regime at the applied voltage of 2.4 V. We observed that oscillations only begin to appear when the voltage is increased beyond 2.7 V. For this study, inducing full oscillation is not necessary and would require longer pulse widths to damp to a 50% switching probability, as noted by the reviewer. This would increase energy consumption, which is undesirable for our applications. To demonstrate the oscillatory behavior, we performed additional measurements using a different batch of devices with lower PMA (while VCMA coefficient remains, the voltage we apply is 1.8 V). In this case, we observed a clear, damped oscillation in switching probability with increasing pulse width, as expected.

Figure R1. Damped oscillation in switching probability for the device with lower PMA.

5. Fig. 2d,e: Shao et al. 3, 87 *Communications Materials* (2022) showed sub-nanosecond precessional VCMA (>100 fJ/Vm) assisted switching of 50 nm MTJs using a voltage lower than 1 V. The authors should benchmark the parameters of their VC-MTJ devices, and implications on the diffusion process and neuromorphic computing performance, against other VC-MTJ reported in the literature such as *Communications Materials* (2022), *AIP Advances* 7, 055934 (2017), *Appl. Phys. Lett.* 108, 012403 (2016), *Appl. Phys. Express* 9, 013001 (2016)

and IEEE Transactions On Nanotechnology 22, 2023, etc.

Response: Thank you for pointing this out. We agree that benchmarking our device against the cited works is important, and we have added these references to the manuscript. We would like to note that several of the works mentioned by the reviewer are from our own research group. Specifically, the study by *Shao et al. (Communications Materials 3, 87 (2022))*, led by Prof. Khalili, was conducted after he moved to Northwestern University. In fact, before that, Prof. Khalili had a long tenure in our group, where he played a key role in developing the VCMA MTJ technology. Given his extensive experience, it is not surprising that he achieved excellent performance in his subsequent work. Our group has demonstrated high VCMA coefficients in the past, including values exceeding 100 fJ/Vm (*Appl. Phys. Lett. 110, 052401 (2017)*). In this study, the VCMA coefficient of 40 fJ/Vm is lower than our previous results. This discrepancy is due to the challenges faced during the integration of the device into CMOS BEOL on an 8-inch wafer, where lab-to-fab transfer issues presented significant hurdles. As this is our first attempt and, to the best of our knowledge, the first full CMOS integration of a VCMA-based MTJ device, we could not yet achieve the sub-volt switching voltage seen in other studies, which require a higher VCMA coefficient.

We remain optimistic that with continued collaboration with the foundry and further optimization of the growth process, the MeRAM performance can be enhanced to match the standards of our single-device lab demonstrations. We agree that achieving this level of performance is crucial for the future application of MeRAM technology. We have modified our wording and added references accordingly.

In Supplementary Note 3

The VCMA coefficient extracted in this work is lower than our previous results and other reported values, which can reach as high as 100 fJ/Vm. We attribute this reduction to challenges in integrating the MTJ stack into the CMOS BEOL process on an 8-inch wafer, which likely affected the interfacial quality and uniformity of the oxide layer. Despite this, the measured coefficient remains sufficient for achieving reliable voltage-controlled switching in our neuromorphic diffusion process, demonstrating the feasibility of our approach even with the current limitations.

6. *Fig:2c-e: Is there any in-plane assistance field applied during measurements? Precessional switching may necessitate the use of an in-plane field.*

Response: Yes, we applied a 390 Oe in-plane field to enable precessional switching, as mentioned in the **Methods** section of the manuscript.

For the switching probability measurements, we use a GMW 5201 Projected Field Electromagnet driven by a Kepco Bipolar Operational Power Supply to apply an external in-plane magnetic field 390 Oe.

7. *Authors should provide a detailed documentation of the pulsing parameters and simulation set up and software in the Method section.*

Response: Thank you for your suggestion. We have moved simulation setup from Supplementary Note 2 to Methods section of the manuscript, and the software we use is MATLAB. Additionally,

the fitting parameters have been included in the Supplementary Information for reference.

We have revised the **Methods** and **Supplementary Information**.

In Method

Simulation of switching probability under different voltage. Macrospin simulations are performed to obtain the switching probability in the thermal activation regime, namely, when the voltage is slightly below the critical voltage that removes the barrier completely. In the simulation, we numerically integrate the Landau-Lifshitz-Gilbert equation using Matlab

$$\dot{\mathbf{m}} = -\gamma_0 \mathbf{m} \times (\mathbf{H}_{eff} + \mathbf{h}) + \alpha \mathbf{m} \times \dot{\mathbf{m}} \quad (3)$$

where the effective field $\mathbf{H}_{eff} = -\partial_{\mathbf{m}} w / \mu_0 M_s$ is obtained from the gradient of the energy density profile w containing the PMA energy and Zeeman energy

$$w = K_{eff}(V)(1 - m_z^2) - \mu_0 M_s H_x m_x \quad (4)$$

The effective anisotropy $K_{eff}(V)$ is controlled by the applied voltage across MgO via the VCMA effect. For sufficiently large K_{eff} , the energy landscape has two energy minima corresponding to the two PMA states. To investigate the switching probability, we include a white noise as the thermal random field \mathbf{h} . Following Brown's derivation,

$$\langle h_i(t) \rangle = 0, \quad \langle h_i(t) h_j(t + \tau) \rangle = \mu \delta_{ij} \delta(\tau) \quad (5)$$

where $\mu = \frac{2k_B T \alpha}{\gamma_0 \mu_0 M_s V}$ is used to satisfy the thermal equilibrium condition. Numerically, we follow Scholz's approach and use the 2nd order Heun's method to integrate the dynamics. Adopting this approach ensures a good balance between numerical stability and complexity.

During the simulation, we initialize the spin in the absence of an external voltage so that the state will evolve from one of the two strong PMA states. Then, we turn on a voltage pulse with sharp edges. The PMA energy term is instantly modified, thereby affecting the dynamics through the effective field. The switching event is determined if the state is trapped in the other PMA state a few nano-second after the voltage is turned off. The switching probability $((P_{AP \rightarrow P} + P_{P \rightarrow AP})/2)$ is obtained by counting the number of the switching events among 150 trials.

In Supplementary Table 4

Parameter	Value
Magnetization M_s [A/m]	10^6
Uniaxial anisotropy K_{u0} [$\text{J} \cdot \text{m}^{-3}$]	7.114×10^5
Damping constant α	0.1
VCAM coefficient ξ [fJ/V · m]	44

MgO thickness t [nm]	1.8
Free layer thickness t_{FM} [nm]	0.95
In-plane magnetic field H_x [Oe]	350
Critical dimension CD [nm]	100

Supplementary Table 4. Simulation parameters for the macrospin model.

8. *Can the author quantify the thermal stability factor or energy barrier for the fabricated devices?*

Response: Thanks for the comments. Here we apply out-of-plane field switching probability method to estimate thermal stability factor (Δ) and effective perpendicular magnetic anisotropy field, $P_{SW}(H_z) = 1 - \exp\left(-\operatorname{erfc}\left(\sqrt{\Delta}\left(1 - \frac{H_z}{H_{keff}}\right)\right)\frac{\sqrt{\pi}H_{keff}}{2R\sqrt{\Delta}}\right)$. The fitted thermal stability is $48k_B T$ while the effective anisotropy field is 1152 Oe. Within single spin approximation, the effective field corresponding to the thermal stability is lower than the fitted value, suggesting that the possible existence of a minimum energy path different from the coherent switching.

We have revised the **Supplementary Information**.

In Supplementary Note 2

Quantify the thermal stability factor. Here we apply out-of-plane field switching probability method to estimate thermal stability factor (Δ) and effective perpendicular magnetic anisotropy field,

$$P_{SW}(H_z) = 1 - \exp\left(-\operatorname{erfc}\left(\sqrt{\Delta}\left(1 - \frac{H_z}{H_{keff}}\right)\right)\frac{\sqrt{\pi}H_{keff}}{2R\sqrt{\Delta}}\right)$$

The fitted thermal stability is $48k_B T$ while the effective anisotropy field is 1152 Oe. Within single spin approximation, the effective field corresponding to the thermal stability is lower than the fitted value, suggesting that the possible existence of a minimum energy path different from the coherent switching.

In Supplementary Figure 2.

Supplementary Figure 2. Fitting of thermal stability. Thermal stability measurement of a MTJ device by sweeping out-of-plane magnetic field.

9. *Is MTJs the optimal arrangement? How does the range and granularity of the noise as a function of number of MTJs affect the FID?*

Response: Thank you for your question. For the first part, we agree that increasing the number of bits in the MeRAM array improves diffusion results. Based on your suggestion, we conducted additional tests, as shown in the following figure. Our results indicate that using 4 to 6 bits can generate images with reasonably good quality for the neuromorphic diffusion process. Additionally, we observed that the energy consumption increases linearly with the number of bits in the array, aligning with our expectations for scalability.

Regarding the impact of noise range and granularity on the FID score, we agree that this is an important consideration. To address this, we conducted additional simulations assuming an idealized MTJ with perfect switching behavior and no variation in switching probability, resulting in an ideal noise distribution function as derived in our theoretical model. We then deployed this idealized noise distribution in the DDPM image generation task and compared the FID scores. Our results showed no significant difference in FID scores compared to those obtained using real MTJ devices, indicating that the range and granularity of the noise in our current implementation are sufficient for the intended application. We believe the reason for this robustness is that the image generation process in DDPM relies on many small, random adjustments rather than a single precise change. Each step of the process involves adding and removing small amounts of noise repeatedly, which helps the model "average out" any small errors. In other words, because the model uses many steps and combines multiple noise samples, any minor inaccuracies or variations in the noise do not have a large impact on the final image quality. This flexibility allows the system to still perform well even if the switching probabilities of our MTJs are not perfectly aligned with the theoretical predictions.

We have revised the **Supplementary Information**.

Supplementary Figure 5. Impact of bit precision and device variability on the image generation quality using MeRAM-based diffusion. **a**, FID score as a function of the number of bits in the MeRAM array, ranging from 2 to 8 bits. The FID score stabilizes beyond 6 bits, and using 4-bit precision already produces relatively good image quality. **b**, Comparison of FID scores for MeRAM-based diffusion with and without device variability. The "ideal" simulation assumes that the switching probability exactly matches the theoretical model, with every MTJ exhibiting identical behavior and no variance. The results show no significant change in FID score, suggesting that the variability of MTJ switching probabilities across devices does not degrade the image generation performance. The reason for this robustness is that the image generation process

in DDPM relies on many small, random adjustments rather than a single precise change. Each step of the process involves adding and removing small amounts of noise repeatedly, which helps the model "average out" any small errors. In other words, because the model uses many steps and combines multiple noise samples, any minor inaccuracies or variations in the noise do not have a large impact on the final image quality. This flexibility allows the system to still perform well even if the switching probabilities of our MTJs are not perfectly aligned with the theoretical predictions.

10. Authors should include a schematic and/or images of the 80x80 VC-MTJ array and circuitry to help readers visualize their setup. Additionally, to show supporting data such as cross-sectional TEM of the integrated MRAM devices on CMOS wafer given the claim that it is the first time it has been achieved.

Response: Thank you for your suggestion. We have added more details in the supplementary material with a cross-sectional TEM image of the integrated MRAM devices on the CMOS wafer, to support our claim of achieving this integration for the first time. Additionally, we have included a schematic of the 80x80 VC-MTJ array and the associated circuitry to help readers better visualize the setup.

Supplementary Figure 7. An 80-by-80 MeRAM array is integrated BEOL (Back End of Line) on a CMOS circuit chip. a, Block diagram of the MeRAM circuit. The MTJs are arranged in an 80-by-80 pattern, and each column has a write driver and a read sense amplifier. All sense amplifiers share a common reference. The timing control generates control signals that transition faster than the digital clock. The chip communicates with a JTAG interface. **b,** Test PCB is situated in a magnet and connected to a microcontroller unit. **c,** TEM image of 1T (Transistor)-1MTJ cell.

11. *The author should comment whether the SOT/STT MTJ, without the VC mechanism, can perform the DDPM. Additional FID simulations should be performed for SOT/STT MTJ, without VC, to substantiate the high-performance image generation capability, if any, of the VC mechanism over SOT/STT-MTJ.*

Response: Thank you for this question. It is possible to perform the DDPM using SOT/STT-based MTJs instead of VCMA MTJs. However, using SOT/STT MTJs would require lowering the energy barrier significantly to achieve the desired tunability for probabilistic bit generation. This low energy barrier would compromise the retention time, making it unsuitable for use as non-volatile memory. In contrast, our VCMA-based MTJ maintains sufficient retention while providing tunable switching probability, allowing it to function effectively as both a p-bit and a memory element. Thus, the VCMA mechanism offers a distinct advantage for neuromorphic applications that require both memory and noise generation capability.

In **Discussion:** (More details are listed in **Supplementary Note 6** and **Supplementary Table 3.**) Furthermore, MeRAM overcomes the limitations of conventional STT-based MRAM, which faces higher energy consumption, limited endurance, and reduced speed due to incubation delay. Additionally, while STT and SOT-MRAM requires a low energy barrier to function as a probabilistic bit (p-bit)—compromising retention time—VCMA-based MTJ devices in offer the unique flexibility to serve as both p-bits and memory cells, making it particularly suitable for large-scale neuromorphic probabilistic computing applications.

Response to Reviewer #3&4

Stochastic diffusion processes, driven by random number sequences, are highly valuable in neuromorphic computing and generative artificial intelligence (AI). However, conventional computers face significant inefficiencies in running AI models based on diffusion processes, primarily due to the separation of storage and computation, as well as the challenges of generating random numbers efficiently. Given the increasing demand for generative AI computing, there is a strong need for new nano-electronic devices that incorporate non-volatility and intrinsic randomness. In this manuscript, the authors present spin-based hardware for Gaussian noise generation and diffusion process. The switching probabilities have been adjusted by varying the voltage pulse width and magnitude in voltage-controlled magnetic tunneling junctions (VC-MTJs) under an external magnetic field. Additionally, they demonstrate image generation tasks using CMOS-integrated voltage-controlled magnetoelectric random access memory (VC-MRAM). While their findings offer a possible approach to efficient Gaussian noise generation and diffusion process, there are some key issues that need to be clarified.

Response: Thank you for your thorough and insightful comments. We appreciate your recognition of the significance of our work and the potential impact on generative AI and neuromorphic

computing. We will address each of your concerns in detail below and have made revisions accordingly in the manuscript. We hope these changes will satisfactorily resolve the issues you raised.

1. Limited tunability of the VC-MTJ switching probability. In Figure 2c, it is unclear why the switching probability does not vary smoothly with the voltage pulse width as expected, and why the $PAP \rightarrow P$ and $PP \rightarrow AP$ are not equal. Moreover, achieving a time resolution as precise as 0.1 ns is challenging in practical circuits, making it difficult to accurately control the switching probability of the VC-MTJ. Do these limitations affect the quality of Gaussian noise generation and the diffusion process? This requires further clarification.

Response: Thank you for this important question. The discrepancy between the switching probability $AP \rightarrow P$ and $P \rightarrow AP$ curves in Figure 2c is most likely due to a residual stray field in the z-direction. While our synthetic antiferromagnetic (SAF) layer was designed to compensate for the stray field, minor fabrication imperfections inevitably introduce some stray field effects. We are actively working on optimizing the growth process and device design to further minimize this issue. Regarding the time resolution, we agree that achieving a precise 0.1 ns resolution is challenging in practical circuits. However, in our noise generation process, we only used pulse widths of 0.4 ns and 2 ns, which are sufficiently distinct from each other and do not require such fine resolution control.

To address the concern by the Reviewer whether these would influence the noise generation and the diffusion process, we conducted additional simulations assuming an idealized MTJ with perfect switching behavior and no variation in switching probability, resulting in an ideal noise distribution function as derived in our theoretical model. We then deployed this idealized noise distribution in the DDPM image generation task and compared the FID scores. Our results showed no significant difference in FID scores compared to those obtained using real MTJ devices, indicating that the granularity of the noise in our current implementation are sufficient for the intended application. We believe this is because the image generation process is naturally quite flexible and can handle some variation or small errors in the noise. The reason for this robustness is that the image generation process in DDPM relies on many small, random adjustments rather than a single precise change. Each step of the process involves adding and removing small amounts of noise repeatedly, which helps the model "average out" any small errors. In other words, because the model uses many steps and combines multiple noise samples, any minor inaccuracies or variations in the noise do not have a large impact on the final image quality. This flexibility allows the system to still perform well even if the switching probabilities of our MTJs are not perfectly aligned with the theoretical predictions.

We have revised the **Supplementary Information**.

Supplementary Figure 5... b, Comparison of FID scores for MeRAM-based diffusion with and without device variability. The "ideal" simulation assumes that the switching probability exactly matches the theoretical model, with every MTJ exhibiting identical behavior and no variance. The results show no significant change in FID score, suggesting that the variability of MTJ switching probabilities across devices does not degrade the image generation performance. The reason for this robustness is that the image generation process in DDPM relies on many small, random adjustments rather than a single precise change. Each step of the process involves adding and removing small amounts of noise repeatedly, which helps the model "average out" any small errors. In other words, because the model uses many steps and combines multiple noise samples, any minor inaccuracies or variations in the noise do not have a large impact on the final image quality. This flexibility allows the system to still perform well even if the switching probabilities of our MTJs are not perfectly aligned with the theoretical predictions.

2. *Uncertainty in the robustness of VC-MTJ probabilistic properties. It remains unclear how many times the VC-MTJ can be reliably controlled by voltage pulses without significant changes in switching probability. The manuscript does not address how the switching probability evolves after repeated application of identical pulses to a single VC-MTJ, which is critical for evaluating the device's lifetime in probabilistic applications. This issue needs to be clarified.*

Response: Thank you for pointing this out. We have reported the endurance and stability of VC-MTJs under repeated voltage pulses in our IEDM 2023 work (Suhail, H., et al. "The First CMOS-Integrated Voltage-Controlled MRAM with 0.7 ns Switching Time." 2023 International Electron Devices Meeting (IEDM). IEEE, 2023), as shown below. R_P or R_{AP} does not change after more than 10^{11} cycles at 1.8V write pulses 0.7ns, indicating good endurance. The differences in switching voltage and resistance compared to earlier batches in this work are due to our continuous refinement of the fabrication recipe with lower PMA.

Figure R2. Measured R_P and R_{AP} from 8 devices. (Suhail, H., et al. IEDM 2023).

Additionally, we have conducted endurance tests to evaluate the robustness of the VC-MTJ’s probabilistic properties on our most recent batch of devices at 1.5 V. The results, shown below, indicate that the switching probability remains stable over extended cycling, with no significant changes observed after 10^{11} pulses. These findings suggest that the VC-MTJ can maintain reliable performance in probabilistic applications, demonstrating the device’s suitability for long-term use.

Figure R3. Endurance of MTJ device. Switching probability (P to AP and AP to P) is obtained after each endurance test.

3. *Device-to-device variability in probabilistic properties.* The manuscript does not address the variability in probabilistic behavior across different devices in the VC-MRAM array. For instance, what is the distribution of switching probabilities within the VC-MRAM array when

identical 2.4V@2ns voltage pulses are applied? If the variability between devices is significant, how does this impact the accuracy of Gaussian noise generation? Furthermore, how should this variability be calibrated to ensure consistent performance?

Response: Thank you for this question. We acknowledge that device-to-device variability in probabilistic properties is an important factor to consider. As mentioned in response to question #1, our experiments were conducted with the variability of real devices taken into account. The generated noise distribution function already reflects the variance and imperfections arising from the device-to-device differences within the MeRAM array. Despite this variability, our tests show that the image quality remains unaffected at the variance level observed in our devices. This is because the diffusion process in DDPM is naturally robust and can accommodate small variations in noise generation without degrading the overall performance, making our MeRAM devices well-suited for this application.

4. *Unclear role of VC-MRAM in denoising diffusion probabilistic models (DDPM). The manuscript does not clearly explain how the noisy dataset generated by VC-MRAM is used during the image generation phase (reverse diffusion process) of the DDPM. How is the switching probability of each VC-MTJ determined in each iteration? It would be helpful to include a detailed explanation of at least one iteration of the specific calculations involved in the reverse diffusion process.*

Response: Thank you for raising this important question. We agree that clarifying the role of VC-MRAM in the reverse diffusion process of the DDPM would help readers better understand our approach. In DDPM, the image generation process starts with pure noise and gradually "reverses" this noise to create a meaningful image. This is done in a series of steps, where at each step the model removes a small amount of noise from the current noisy image, bringing it closer to the final, denoised image. This process is called reverse diffusion, and it relies heavily on high-quality random noise samples to guide the removal of noise correctly at each step. In our demonstration as shown in Fig. 3c, the image is represented by an 80×80 array of MTJs. Each pixel corresponds to the state of an MTJ: dark blue dots indicate the parallel (P) state, which we represent as "1," and light blue dots indicate the antiparallel (AP) state, represented as "0." During the reverse diffusion process, the goal is to iteratively "denoise" the image, moving from a completely noisy state back towards a clear image. The MeRAM generates random noise samples at each step by controlling the switching probabilities of the MTJs using voltage pulses. The noise is expressed as small changes in the coordinates of the pixels from the previous step, guiding the image towards a denoised state in a controlled manner. This controlled noise injection helps the model determine the amount of noise to subtract from the image in the next step. Thus, we can say the switching probability of each VC-MTJ is determined by the model based on previous states of our image (80 by 80 MTJs). The ability to adjust the switching probability of the MTJs allows us to tune the generated noise, matching the noise schedule required by the diffusion process. In simpler terms, MeRAM creates the specific type of noise needed by the model at each step, enabling the image to be "cleaned" gradually. This approach improves the efficiency of the image generation process by leveraging the intrinsic randomness and tunability of VC-MTJs, reducing the need for slower, software-based noise generation.

We have added a more detailed explanation of this process in the revised manuscript.

In **Supplementary Note 7**

In our demonstration of image generation tasks (As shown in Fig. 3), we use an 80×80 VC-MTJs to represent the image data. Each pixel corresponds to an individual MTJ, where dark blue dots indicate the P state (1) and light blue dots indicate the AP state (0). Instead of modifying the pixel intensity values as in the standard DDPM due to the limited number of devices, our hardware-based approach uses the noise generated by the MeRAM array to adjust the coordinates of the P state pixels (dark blue dots), effectively altering the visual appearance of the image. During the reverse diffusion process, the role of the MeRAM is to produce controlled noise samples, which are used to update the state of each MTJ based on the applied voltage pulses. The noise in this context is reflected as changes in the coordinates of the P state pixels between consecutive steps, simulating the reverse diffusion process. This adjustment guides the image gradually towards a clearer representation by iteratively "subtracting" the noise introduced during the forward diffusion. By leveraging the tunable switching probabilities of the VC-MTJ, we are able to generate noise samples required by the diffusion model.

5. *Requirement for an external magnetic field. The need for an external magnetic field is not conducive to practical applications, as its high energy consumption undermines the goal of low-power applications. Additionally, it raises the question of whether a slanted magnetic field affects the switching probability of the VC-MTJ. This should be addressed.*

Response: Thank you for raising this important point. It is true that, in this work and typically in precessional VCMA switching, an in-plane magnetic field is required. The in-plane field helps maintain the single-domain state of the magnetic moment when the VCMA effect lowers the energy barrier, enabling stable precessional switching. We agree that the need for an external field is not ideal for practical, low-power applications. However, the community has proposed integrating magnetic hard mask layer on top of the MTJ to provide a in-plane stray field, which could replace the need for an external magnetic field and make the system more energy efficient (Garello, Kevin, et al. "Manufacturable 300mm platform solution for field-free switching SOT-MRAM." 2019 Symposium on VLSI Circuits. IEEE, 2019.)

We are also working on addressing this issue. In our ongoing research, we have made significant progress in achieving VCMA switching without the need for an external magnetic field. We found that the Dzyaloshinskii-Moriya interaction (DMI) can help stabilize the magnetic moment even in the absence of an applied field. Here, we show some of our preliminary results demonstrating this effect. These results are not yet published, so we kindly ask for confidentiality regarding this information.

Figure R4. Field free VCMA switching. Switching probability as a function of pulse width at 1.7 V without external magnetic field.

Minor issues:

6. *What is the critical switching voltage for spin-transfer torque (STT)-induced free-layer magnetization switching in the presence of in-plane magnetic fields? Additionally, why is the contribution of STT omitted in device simulations?*

Response: Thank you for your question. In the case of our VCMA MTJ, we require an applied electric field rather than a direct current for switching. As shown in our data, the TMR is on the order of hundreds of $k\Omega$, resulting in a leakage current density of only $\sim 5 \times 10^8$ A/m². This is significantly lower than the typical current density required for STT switching, which is on the order of $\sim 1 \times 10^{11}$ A/m². Achieving STT switching usually requires MTJ with TMR values in the a few $k\Omega$ range. Thus, we can omit the STT effect in our modeling, as it does not play a significant role under the conditions used in our experiments.

7. *In Figure 2a, why does the data point at 500 mV/nm appear significantly offset from the fitted line?*

Response: Thank you for your careful reading of the data. The offset of the data point at 500 mV/nm from the fitted line in Figure 2a is indeed noted. We suspect that the large DC voltage applied during the calibration of the VCMA coefficient may have induced some localized heating, leading to a reduction in the PMA energy (K_u) and causing the observed offset. Overall, the linear fit captures the trend well and provides a reliable extraction of the VCMA coefficient.

8. *The widths of the input voltage pulses applied to the 8-bit VC-MTJ array shown in Figure 3a do not correspond to those described on page 8, line 3-4 of the main text.*

Response: Thank you for pointing this out. We are sorry for the typo. We have corrected Fig. 3a.

9. *The formula for A_i in the penultimate line of the second paragraph on page 7 is incorrect.*

Response: Thank you for pointing this out. It appears that this issue may have been caused by a bug during the PDF conversion in the Nature submission system, as our local PDF shows the formula correctly. We will double-check and ensure that the formula is displayed correctly in the revised submission.

10. Recently, non-volatile field-free spin-orbit torque (SOT) devices have been demonstrated for probabilistic bits. These studies should be considered when comparing VC-MRAMs with SOT-MRAMs: [Nano Lett. 24, 5420-5428 (2024); Nano Lett. 24, 10072-10080 (2024)].

Response: Thank you for pointing out these important works. We have added these references to our manuscript and appreciate your suggestion. And we would like to point out that although it is possible to use SOT-based MTJs as p-bit. However, it requires lowering the energy barrier significantly to achieve the desired tunability for probabilistic bit generation. This low energy barrier would compromise the retention time (thermal stability), making it unsuitable for use as non-volatile memory. In contrast, our VCMA-based MTJ maintains sufficient retention while providing tunable switching probability, allowing it to function effectively as both a p-bit and a memory element. Thus, the VCMA mechanism offers a distinct advantage for neuromorphic applications that require both memory and noise generation capability.

In Discussion:

Furthermore, MeRAM overcomes the limitations of conventional STT-based MRAM, which faces higher energy consumption, limited endurance, and reduced speed due to incubation delay. Additionally, while STT and SOT-MRAM requires a low energy barrier to function as a probabilistic bit (p-bit)—compromising retention time—VCMA-based MTJ devices in offer the unique flexibility to serve as both p-bits and memory cells, making it particularly suitable for large-scale neuromorphic probabilistic computing applications.

Response to Reviews on Manuscript NCOMMS-24-53635A

We thank the four Reviewers for their careful and prompt reviews. Below, we address the Reviewers' questions and comments on a point-by-point basis. We also revised the manuscript and highlighted the changes in blue.

Response to Reviewer #1

The authors address most of my comments and expand their work accordingly. Although the device development part is not accurate as the diffusion method, I think that one of the value of their paper is the implementation of the computing system. I recommend their work for publication in Nature Communications

Response: Thank you for your positive feedback and for recommending our manuscript for publication in *Nature Communications*. We appreciate your acknowledgment of our work's value in implementing the computing system. We also recognize your concern about the device development part and will continue refining our approach to address any limitations in our experimental setup. Your constructive review has helped improve our work, and we are grateful for your support.

Response to Reviewer #2

The authors have addressed most of the questions satisfactorily by and large and we deem the manuscript to be technically sound. However, there is still a concern on the novelty aspect of this work. The concept of stochastic/probabilistic computing using MRAMs have been well demonstrated (10.1109/NANO61778.2024.10628878, 10.1109/JXCDC.2022.3231550), wherein SOT-MRAM and voltage assisted/MeRAM have been shown to be similarly suited for these systems with small delay and low energy. The use of MeRAM for probabilistic computing is not new either (10.1088/1361-6528/acf6c7, 10.1109/NANO61778.2024.10628878). In this regard, we feel that the significance of this work is not strong and tends towards incremental.

Response: Thank you for your detailed comments and for pointing out the previously demonstrated work on stochastic/probabilistic computing using MRAMs. We acknowledge that the concept of probabilistic computing is not new, and we will include references to those important prior studies in our revised manuscript. However, our current work extends beyond demonstration at the few-device level by utilizing a fully integrated CMOS-based MeRAM chip for diffusion processes and generative AI models. The application of this technology for diffusion and AI-related tasks, with a system-level demonstration, distinguishes our study from existing literature. We believe this integrated approach contributes novel insights and furthers the applicability of spin-based hardware in advanced computing contexts.

Response to Reviewer #3&4

I believe the manuscript has been significantly improved with these clarifications and additions.

The authors have addressed the key concerns raised in my review, and I now feel that the manuscript provides a more comprehensive understanding of the proposed spin-based hardware for Gaussian noise generation and diffusion processes. I am confident that the revisions will enhance the quality and applicability of this work. I am happy with the current format of the paper.

Response: Thank you for your positive feedback and for recognizing the improvements in our revised manuscript. We are pleased that you find the current format satisfactory, and we value your support and suggestions of our work.